



# Predicting decadal trends in cloud droplet number concentration using reanalysis and satellite data

Daniel T. McCoy[1*], Frida A.-M. Bender[2], Daniel P. Grosvenor[1], Johannes K. Mohrmann[3], Dennis L. Hartmann[3], Robert Wood[3], Paul R. Field[1,4]

[1]University of Leeds, Leeds, LS2 9JT, UK, [2]Stockholm University, Stockholm, 114 18, Sweden
[3]University of Washington, Seattle, 98195, USA [4]MetOffice, Exeter, EX1 3PB, UK

*Correspondence to*: Daniel T. McCoy (d.t.mccoy@leeds.ac.uk)

**Abstract.** Cloud droplet number concentration (CDNC) is the key state variable that moderates the relationship between aerosol and the radiative forcing arising from aerosol-cloud interactions. Uncertainty related to the effect of anthropogenic aerosol on cloud properties represents the largest uncertainty in total anthropogenic radiative forcing. Here we show that regionally-averaged time series of Moderate-Resolution Imaging Spectroradiometer (MODIS) observed CDNC are well-predicted by MERRA2 reanalysis near-surface sulfate mass concentration over decadel timescales. A multiple linear regression between MERRA2 reanalysis masses of sulfate ($SO_4$), black carbon (BC), organic carbon (OC), sea salt (SS), and dust (DU) shows that CDNC across many different regimes can be reproduced by a simple power law fit to near-surface $SO_4$, with smaller contributions from BC, OC, SS, and DU. This confirms previous work using a less-sophisticated retrieval of CDNC at monthly time scales. The analysis is supported by examination of remotely-sensed sulfur dioxide ($SO_2$) over maritime volcanoes and the east coasts of North America and Asia, revealing that maritime CDNC responds to changes in $SO_2$ as observed by the Ozone Monitoring Instrument (OMI). This investigation of aerosol reanalysis and top-down remote sensing observations reveals that emission controls in Asia and North America have decreased CDNC in their maritime outflow on a decadal time scale.

## 1. Introduction

The degree to which anthropogenic aerosol has affected the Earth's albedo by altering cloud properties remains the largest uncertainty in our understanding of how much humans have changed the top of atmosphere energy balance, and by extension what the observed trend in surface temperature tells us about the climate's sensitivity to perturbation (Forster, 2016;Boucher et al., 2014). Aerosol indirect effects can be grouped into two categories: the first indirect effect, or Twomey effect (Twomey, 1977), by which enhanced concentrations of cloud condensation nuclei (CCN) enhance CDNC (for a fixed liquid water content), leading to an increase in cloud albedo; and the lifetime, or Albrecht effect (Albrecht, 1989), by which enhanced CDNC suppresses precipitation and leads to thicker or more persistent clouds and higher cloud albedo. The first indirect effect has been supported by numerous empirical studies relating remotely-sensed aerosol properties to remotely-sensed CDNC (Bellouin et al., 2013;Gryspeerdt et al., 2016;Patel et al., 2017;Quaas et al., 2008;Quaas et al., 2009;Matsui et



al., 2006;Nakajima et al., 2001;Sekiguchi et al., 2003), although whether aerosol affects cloud lifetime is still debated (McCoy et al., 2017b;Malavelle et al., 2017;Gryspeerdt et al., 2016). Studies have utilized the natural laboratory provided by transient degassing volcanoes to study cloud responses to changes in aerosol (Mace and Abernathy, 2016;Gassó, 2008;Yuan et al., 2011;McCoy et al., 2017b;Malavelle et al., 2017). In this vein, McCoy et al. (2017a) used aerosol reanalysis to provide additional information regarding aerosol speciation and vertical structure. They found that monthly-mean CDNC and sulfate mass concentration near the surface were linked by a power law relationship that remained robust across different regions with very different aerosol properties and cloud regimes, but their analysis was hampered by remote-sensing bias leading to different regions having a different constant term in the log-log fit between CDNC and sulfate ($SO_4$). This study utilizes a new CDNC data set filtered for retrieval error that rectifies these biases. We show that the power law relationship between sulfate and CDNC applies across all regimes. Further, we show that long-term trends in observed sulfur dioxide ($SO_2$) and reanalysis $SO_4$ predict trends in CDNC, indicating that changes in sulfur have the ability to influence CDNC on an inter-annual timescale that is of relevance to the aerosol-cloud radiative forcing.

## 2. Methods

The analysis performed in this study parallels the analysis in McCoy et al. (2017a). Here a much more refined data set is used to analyze the period 2003-2015 (as opposed to 2001-2013 in McCoy et al. (2017a)), expanded to a daily time scale over the entire globe (21,379,174 daily 1°x1° observations). Aerosol reanalysis from MERRA2 is used to gain insight into speciation and vertical distribution that is not provided by remote-sensing analyses that use column-integrated CCN proxies such as aerosol index (AI) or aerosol optical depth (AOD). It has been demonstrated that model-simulated AI accurately predicts changes in CDNC, in contrast to AOD (Gryspeerdt et al., 2017), but observations of AI are still subject to near-cloud retrieval artefacts (Christensen et al., 2017). The aerosol species considered in the present analysis are dust (DU), sea salt (SS), black carbon (BC), organic carbon (OC) and sulfate ($SO_4$). As in McCoy et al. (2017a), DU and SS masses as predicted by MERRA2 are restricted to submicron sizes because these will be more numerous as CCN (Ghan et al., 1998). Similarly, only hydrophilic BC and OC as predicted by MERRA2 are considered. The daily-mean near-surface (~1km altitude) mass concentrations of all aerosol species are calculated by averaging the 3-hourly aerosol mass concentration at the 910 hPa model level in MERRA2 resolved at 0.5°x0.626° resolution to 1°x1° and daily resolution.

In McCoy et al. (2017a) 1°x1° daily-mean MODIS effective radius ($r_e$) and optical depth ($\tau$) values were used to calculate CDNC, which was then averaged to monthly resolution. The use of this CDNC dataset may be problematic in some regions for a number of reasons (also see McCoy et al. (2017a)) : 1) it is subject to high solar zenith angle biases in the individual swaths, which were averaged together to create each daily data point; 2) biases may be present due to the use of area averaged $r_e$ and $\tau$ rather than using pixel level values for the CDNC calculation; 3) the dataset was not filtered to include low altitude clouds only, which may have led to a lack of connectivity between surface aerosol sources and cloud CDNC; 4) the CDNC was calculated using the 2.1μm MODIS channel $r_e$, which is likely to be affected more strongly by cloud





heterogeneity related biases than the 3.7μm channel (Zhang et al., 2012). In the present study, level-2 swath data (joint product) from MODIS collection 5.1 (King et al., 2003) is filtered to remove problematic retrievals at a pixel-level following Grosvenor and Wood (2014), including the removal of pixels with a solar zenith angle greater than 65°. The daily-mean CDNC at 1°x1° resolution is calculated using filtered level 2 swath data and only low (cloud tops below 3.2 km), liquid

clouds were used to calculate CDNC. Only 1°x1° regions where the cloud fraction exceeds 80% are considered valid (Bennartz et al., 2011) and the CDNC is calculated using the 3.7μm MODIS channel $r_e$. The mean CDNC over the period 2003-2015 is shown in Figure 1. Values of CDNC are in-cloud.

The CDNC retrieval from MODIS and the aerosol reanalysis are independent data sets. MODIS AOD is used to nudge MERRA2 reanalysis, where AOD is corrected for near-cloud aerosol swelling (Rienecker et al., 2011;Randles et al.,

2016). However, to develop the usefulness of MODIS CDNC as a measure of aerosol-cloud interactions and the microphysical state of liquid-topped clouds, we utilize in-situ aircraft measurements of CDNC and the sulfur dioxide ($SO_2$) retrieved by the ozone monitoring instrument (OMI). The data set used in this study to examine changes in $SO_2$ is the planetary boundary layer (PBL) $SO_2$ calculated using principal component analysis to reduce artefacts and noise (Li et al., 2013). The retrieval requires a clear-sky, making the $SO_2$ retrievals non-coincident with CDNC retrievals. However, in this

study $SO_2$ is only considered on a regional scale, as opposed to attempting to co-locate it with CDNC data, and so the locally non-coincident nature of these retrievals is not an issue.

We evaluated both volcanic point sources in relatively pristine maritime regions (Carn et al., 2017) and the emissions from Asia and North America (Krotkov et al., 2016). Volcanic plumes and anthropogenic emissions produce very different $SO_2$ signatures, and large volcanic eruptions need to be removed to examine the effect of anthropogenic sources

(Krotkov et al., 2016). The 2008 eruption of Kasatochi emitted a large quantity of sulfur dioxide near 10-12km altitude (Krotkov et al., 2010), rendering the data from August 2008 over the US east coast spurious in terms of examining the trend in anthropogenic sulfur emissions. This has been noted in previous studies (Krotkov et al., 2016) and all data for August 2008 has been removed from analysis of the long-term trend in this region.

It is important to evaluate whether MODIS CDNC offers a useful measure of the real mean CDNC for which in-situ

observations are likely to provide an accurate proxy. We take a different tack from previous studies whose goal was to evaluate whether MODIS CDNC is reliable on a pixel-by-pixel basis (Painemal and Zuidema, 2011;Bennartz and Rausch, 2017). Bennartz and Rausch (2017) showed that their CDNC data averaged over both ~0.2x0.2° and 0.5x0.5° regions correlated strongly with airborne observations from the VOCALS-REX campaign (Painemal and Zuidema, 2011). Here we reprise the analysis in Bretherton et al. (2010) and examine whether the average MODIS retrieval from the sampled cloud

population is similar to an average aircraft observation. Aircraft measurements are taken from literature sources detailing systematic transects across regions with liquid-topped cloud (Lachlan-Cope et al., 2016;Ma et al., 2010;Hegg et al., 2007;Allen et al., 2011;Lu et al., 2007). The flight-leg mean CDNC reported by each study is compared to the relevant MODIS CDNC. Because MODIS CDNC retrievals that are considered reliable by our methodology can be quite sparse, an





average of the region within ±1.5° of the mean location of the flight-leg and one day before and after is taken to calculate the mean CDNC that MODIS would equivalently measure during the flight leg.

Finally, in this study we subdivide our global data set into sub-regions to show sensitivity to sample. These regions are similar to the regions defined in McCoy et al. (2017a) and are shown in Figure 1. Latitude and longitude ranges are given

in Table 1.

## 3. Results

In this section, we evaluate how closely aircraft and satellite measurements match each other in keeping with previous studies (Bretherton et al., 2010;Painemal and Zuidema, 2011;Bennartz and Rausch, 2017). We also examine how

much daily variability in aerosol species influences CDNC; and how this variability is able to predict trends and interannual variability in observed CDNC.

### 3.1 Comparison of in-situ and observed CDNC

First, we establish whether our CDNC concentration data set is consistent with in-situ measurements. To evaluate the CDNC observations from our data set we compare to aircraft observations over a wide range of different regimes. Data

from aircraft campaigns were taken from published literature values detailing mean CDNC for individual flight legs. The idea underlying this methodology is that if the aircraft and MODIS are both measuring the same population, then their mean CDNC values should agree, assuming that both the aircraft and MODIS are sampling randomly. Aircraft measurements from the Antarctic Peninsula, Northern China, and the Peruvian and Californian stratocumulus decks are compared to MODIS CDNC (Figure 2). The correlation between aircraft and satellite observations is r=0.68. This result is very near to the

correlation found by Bretherton et al. (2010) using only the VOCALS-REX data, although it is worth noting that the substantial number of in-cloud transects from that study significantly contribute to the weight of data examined here. When the data from all of the flight legs are binned together most of the bin mean CDNCs from aircraft and remote-sensing observations agree within the standard error in the estimation of the bin means ($\sigma/\sqrt{n}$) (Figure 2). It is important to note that this analysis is intended to illustrate that the CDNC measured by aircraft and the CDNC observed by MODIS are not

drawing from entirely different populations and that the correlation between flight leg CDNC and remotely sensed CDNC is similar to previous analysis (Bretherton et al., 2010). A more rigorous analysis of aircraft and the MODIS CDNC dataset shown in this paper will be undertaken in a future work pending the compilation of a dataset of aircraft CDNC in the spirit of the Global Aerosol Synthesis and Science Project (GASSP) (Reddington et al., 2017).



### 3.2 Covariability between observations of daily CDNC and MERRA2 aerosol mass

We have just shown that the daily-mean CDNC that MODIS observes is consistent on average with in situ observations. How then does this satellite retrieval covary with aerosol mass concentrations on a global scale? We split our global dataset into many different regions selected to focus on either particular cloud regimes or different aerosol emission sources (see discussion in McCoy et al. (2017a)). If the dependence of CDNC on aerosol is similar across these regions, then it supports the idea that this relationship is mechanistic.

We find that as in previous studies, CDNC is strongly dependent on sulfate mass (Figure 3a) and this dependence is similar across many of the regions shown in Figure 1. Notable exceptions are the North Pacific midlatitudes and Indian subcontinent. The former may relate to the challenge presented to reanalysis in predicting daily 1°x1° sulfate mass concentration after advection from Asia across the Pacific; the Indian subcontinent may represent a region where substantial emissions of carbonaceous species render variability in sulfate less important, or it may relate to retrieval difficulties in distinguishing cloud from haze over the subcontinent (Ramanathan et al., 2001). We fit the following regression model

$$\log_{10} CDNC = a_1 \log_{10}(SO4) + a_2 \log_{10}(BC) + a_3 \log_{10}(OC) + a_4 \log_{10}(SS) + a_5 \log_{10}(DU) + b \qquad (1)$$

which differs from the previous study (McCoy et al., 2017a) by adding organic carbon as a predictor. Several of the predictors co-vary strongly (Figure S1). We attempt to ameliorate the issue of collinearity by training separate regression models in each of the regions shown in Figure 1. For example, the correlation between BC and $SO_4$ will be high in regions with significant biomass burning, but non-existent in the remote Southern Ocean. If the regression coefficient relating CDNC to $SO_4$ remains consistent between these regions, then it is a good indication that this relationship is robust.

The coefficients from the multiple linear regression model trained in each of the areas shown in Figure 1 are shown in Figure 3b. Because some aerosol species have little to no variability, the value of each coefficient is shown scaled by the standard deviation over all observations from 2003-2015 of the relevant term in the regression model. Correlations and unscaled regression model coefficients for each region are given in Table 1. If we only train the regression model using daily-mean data from stratocumulus decks, then the coefficient relating $\log_{10}$ sulfate to $\log_{10}$ CDNC remains approximately unchanged relative to McCoy et al. (2017a) (Figure 3b, and Table 1), supporting the estimate by McCoy et al. (2017a) that the increase of CDNC caused by sulfate results in a radiative forcing of -0.97 Wm$^{-2}$. Overall, it appears that daily aerosol reanalysis has the ability to predict day-to-day variations in observed CDNC with a remarkably high degree of skill.

One surprising result from this analysis is the weakly-negative to near-zero dependence of CDNC on submicron sea salt mass. Sea salt is plentiful and hygroscopic and it would seem reasonable to suspect that it would significantly affect CDNC. Analysis of the dependence of CDNC on sea salt and sulfate shows that sea salt mass is only important for very low sea salt mass (Figure 4, for values of $\log_{10}(SS)$ less than roughly -3 increasing sea salt increases CDNC). Presumably this indicates that in situations where sea salt emissions are low it has a limiting effect on the creation of CCN. However, the





effect of sea salt emissions on CDNC appears to be saturated for the majority of observations with increasing sea salt slightly decreasing CDNC (Figure 4, the distribution of MERRA2 sea salt mass over oceans is shown as white contours). This is why the linear regression model assigns it a weakly negative coefficient (Figure 3b). This reduction in CDNC for increasing SS mass may be consistent with large sea salt particles reducing the supersaturation, resulting in fewer accumulation mode

aerosol being activated (Ghan et al., 1998). It is also possible that submicron sea-salt aerosol number does not scale with mass. We have constrained the sea salt mass concentration to only include submicron sea-spray in an attempt to consider only the most CCN-relevant aerosol. However, the MERRA2 reanalysis simply uses wind speed and SST to predict sea spray flux based on a parameterization (Gong, 2003;Jaeglé et al., 2011) and in the context of the analysis presented in this paper the relation between submicron sea salt mass and CDNC is at some level the relation between near-surface wind speed

and CDNC. The precise values of the coefficient should change if a different size distribution is used in the parameterization, but it is likely that the qualitative dependence of CDNC on sea salt would remain the same.

       Another interesting note is the negative dependence of CDNC on BC. This appears to only be a feature of low BC and OC load (Figure S2), but this may also reflect existing issues in the MERRA2 reanalysis of carbonaceous species in terms of representation of aerosol index and vertical distribution in relation to organic carbon (Randles et al., 2016). It is

worth pointing out, however, that there are a priori physical reasons to expect black carbon to thin cloud cover via the semi-direct effect (Hansen et al., 1997).

       As we have seen CDNC covaries substantially with aerosol on a daily scale over the period 2003-2015 and across many different regimes. In particular, we find that sulfate aerosol covaries strongly with CDNC, which is consistent with pioneering work examining cloud-aerosol interactions (Charlson et al., 1992). Our study provides the first systematic top-

down estimate of this covariability.

### 3.3 Decadal trends in CDNC driven by sulfur fluxes

       While our results are consistent with previous work regarding aerosol-cloud indirect effects, it is important to demonstrate that the sulfate-CDNC correlation is not spuriously created by e.g., advection of pollution sources being correlated with meteorological conditions that lead to high CDNC. It is also important to show predictive capability over the

timescales of years and decades that is useful for understanding the radiative forcing from the aerosol-cloud interactions during the industrial era. One way to demonstrate this is by examining known sources of sulfate whose emission flux is unrelated to seasonal or meteorological variability (in contrast to biogenic sulfate, for example).

       For the analysis presented in this paper we adopt the technique used in previous studies (Gassó, 2008;Mace and Abernathy, 2016;Yuan et al., 2011;McCoy and Hartmann, 2015;McCoy et al., 2017b;Malavelle et al., 2017) and examine the

response of cloud properties to volcanic sulfate sources. We support this analysis by examining the systematic change in anthropogenic sulfur emissions from Asia and North America due to emissions controls (Krotkov et al., 2016), as in previous studies (Bennartz et al., 2011), although our data record extends over a period of enhanced emissions controls in East Asia and thus we anticipate a decrease in CDNC in contrast to Bennartz et al. (2011). We examine systematic changes in CDNC





in maritime regions where there is outflow from anthropogenic pollution sources because McCoy et al. (2017a) inferred a strong aerosol-cloud radiative forcing in such regions based on a power law relationship between sulfate and CDNC. Such a long-range relationship between sulfur sources and CDNC would be supportive of sulfate driving CDNC variability.

In the analysis presented below we will examine long-term trends in CDNC as observed by MODIS and predicted by MERRA2 sulfate mass. The notion that these long-term trends originate from changes in sulfur flux from volcanism or emissions controls will be supported by analysis of the boundary-layer $SO_2$ detected by the OMI instrument, which is an independent data set to either reanalysis sulfate mass or MODIS CDNC. Days where data over each region are missing from the time series (for example August 2008 over North America from OMI(Krotkov et al., 2010)) are filled by linear interpolation before applying a 365-day moving average. To allow ease of comparison to trends in $log_{10}$ sulfate mass, $log_{10}$ CDNC is shown in Figure 5. To our knowledge this is the first study to show that variations in anthropogenic emissions drive changes in CDNC using remotely-sensed $SO_2$ and CDNC.

The volcanoes on the Pacific islands of Vanuatu and Hawaii constitute the largest volcanic sources of sulfur dioxide in the data record afforded by OMI (Carn et al., 2017). Their relatively pristine remote locations and large inter-annual variability in sulfur emissions make them ideal for examining covariation between CDNC and PBL $SO_2$. The average CDNC and $SO_2$ within 5° of the volcanoes is shown during the period 2003-2015 (Figure 5a,b).

The variances in daily PBL $SO_2$ detected by OMI and in CDNC detected by MODIS are correlated in the vicinity of both Vanuatu and Hawaii (Figure 5a,b). Increased volcanic activity observed in-situ at Kilauea in Hawaii during 2008-2010 (Elias and Sutton, 2012;Longo et al., 2010) translates to a strong increase in $SO_2$ as measured by OMI and in CDNC as measured by MODIS, with a nearly four standard deviation increase in CDNC and $SO_2$ at its peak. The activity near Vanuatu is less pronounced, but the MODIS-observed CDNC still covaries with long-term trends in OMI $SO_2$. These results suggest that variability in CDNC on the time scales of months and years is being driven by volcanism in these regions.

Volcanic sources play an important role in determining pre-industrial CDNC (Schmidt et al., 2012), but one of the central goals of the analysis presented in this work is to offer a constraint on CDNC changes due to anthropogenic activity. Emissions controls in both China and the United States have resulted in steadily decreased $SO_2$ emissions in these regions over the observational record from OMI (Krotkov et al., 2016). The $SO_2$ measured over land on the east coast of North America (30°-45°N,85°-65W) and Asia (10°-40°N,110°-120°E) is shown in Figure 5c,d. This decrease in $SO_2$ over continents correlates well with CDNC observed over the Pacific (10°-40°N, 110°-150°E) and Atlantic (30°-45°N, 80°-65°W) (Figure 5c,d). Land domains were chosen to match the regions of $SO_2$ production in China and the US examined in Krotkov et al. (2016). As noted in Krotkov et al. (2016), the Yangtze River delta, Pearl River delta, and Sichuan Basin contribute the majority of emissions in China, while Pennsylvania and the Ohio River valley contribute strongly to North American emissions. The averages over land have been selected to capture these regions and agree with previous studies(Krotkov et al., 2016). The accompanying maritime outflow regions have been chosen to match the same latitude range and capture the region of enhanced CDNC shown in Figure 1.





It is interesting to note that the trends in $SO_2$ over Asia and North America and related CDNC changes over the Pacific and Atlantic parallel the history of emissions controls in China and the United States (US), supporting the idea that the observed trend is related to aerosol affecting cloud properties, as opposed to some systematic change in circulation during the observational record. In the US, various federal and state-level controls on sulfur emissions such as the 1990

Clean Air Act, the 2010 Acid Rain Program, and the 2009 Clean Air Interstate Rule have led to a steady decrease in $SO_2$ over the US east coast. This trend appears in OMI observations and is corroborated by ground-based and aircraft inventories (Krotkov et al., 2016;He et al., 2016;Hand et al., 2012).

Sulfur dioxide over China does not exhibit as monotonic a behavior as the east coast of North America. $SO_2$ decreases substantially during the period 2008-2010, which has been suggested to result from a combination of economic

recession and the emission control measures put in place before the 2008 Olympic Games in Beijing (Krotkov et al., 2016;Li et al., 2010;Lu et al., 2011;Mijling et al., 2009;Witte et al., 2009). CDNC over the Pacific decreases during this period as well, although not for as long as $SO_2$ (Figure 5c). Since 2012 $SO_2$ over eastern China has decreased substantially. This may reflect emission controls implemented as part of the 12[th] five-year plan (Tian et al., 2013;Zhao et al., 2013), as well as cleaner coal-fired technology (Wang et al., 2015).  The strong decrease in $SO_2$ from 2012 is mirrored in trends in CDNC

over the Pacific. Taken together, these long-term trends in maritime CDNC responding to continental emissions of sulfur dioxide underline the link between sulfate and CDNC.

In addition to the strong pollution sources in North American east coast and East Asia, we also investigated inter-annual variability associated with the European Union and the stratocumulus decks listed in Table 1. Interannual variability in these regions is less dramatic and $SO_2$ is generally below the OMI detection threshold (Krotkov et al., 2016), making

interpretation of the long-term trends in $SO_2$ difficult. However, in the European Union, and Peruvian stratocumulus regions variability in CDNC and $SO_2$ agree moderately well (Figure S3 and Figure S4).

We have examined the covariability between remotely-sensed PBL $SO_2$ and CDNC. In both pristine and polluted regions, long-term trends in CDNC appear to be driven by changes in sulfur flux (Figure 5). This leaves us with an important question for this analysis: how well does the sulfate mass from MERRA2 replicate these decadal trends? The long-term

trends in $\log_{10}$ CDNC are well-correlated with long-term trends in $\log_{10}$ $SO_4$, with the notable exception of the Australian stratocumulus region (Figure S3, Figure 5, Figure 6). This is probably because this region is dominated by biogenic sulfur produced by marine organisms (McCoy et al., 2015;Rap et al., 2013;Kloster et al., 2006). The MERRA2 reanalysis uses a climatology to inform it about fluxes of dimethyl-sulfide (Randles et al., 2016) and it has very limited ability to simulate inter-annual variability. Note that correlations provided in Figure 5 are between unsmoothed time series. The correlation

between time series treated with a 365-day running mean are provided in Figure 6.

It is interesting to examine how well our predictions of the sensitivity of CDNC to $SO_4$ based on daily variability extend to long-term trends. The coefficient linking $\log_{10}$ $SO_4$ to $\log_{10}$ CDNC inferred from 1°x1° daily data in the stratocumulus regions agrees with the relation between inter-annual variations in $\log_{10}$ $SO_4$ and $\log_{10}$ CDNC. This is shown in Figure 6. The range of coefficients arrived at by training the regression model in the stratocumulus regions (Table 1)





captures the coefficients linking inter-annual variations in $\log_{10}$ SO$_4$ to $\log_{10}$ CDNC (Figure 6). Most regions appear to be closer to the regression model trained in Australian stratocumulus, with the exception of the Hawaiian and Californian regions, which are closer to the regression model trained in Californian stratocumulus. It should be noted that the decadal trends in CDNC and SO$_4$ shown in Figure 5 are not driving the training of the regression model because the variance in

1°×1° daily-mean CDNC exceeds the variance in regional-mean CDNC by almost three orders of magnitude after the application of the 365-day moving average. Overall, it appears that the regression models trained in the stratocumulus regimes using daily data have the capability of predicting long-term variability in a variety of different regimes.

One interesting aspect of this analysis is that, while the time series of observed and predicted CDNC are well-correlated (see Figure 5 for correlation between unsmoothed time series and Figure 6 for correlations between time series

after the application of a 365-day running mean), uncertainty still exists in the sensitivity of CDNC to SO$_4$ as characterized by the coefficient relating CDNC to SO4 in Equation 1. It is unclear if this diversity is due to a real difference in the way that clouds and aerosol interact in these regions, perhaps due to differences in the effects of nucleation on CCN concentration (Gordon et al., 2016;Dunne et al., 2016), or if it is due to shortcomings in reanalysis or retrievals.

## 4. Conclusions

Several studies have shown that sulfate mass concentration influences CDNC (Boucher and Lohmann, 1995;Lowenthal et al., 2004;McCoy et al., 2017a;McCoy et al., 2015;Storelvmo et al., 2009). Previous studies relating sulfate mass to remotely-sensed CDNC were hampered by significant retrieval bias, making the interpretation of their results difficult (McCoy et al., 2017a). In this study we utilize daily-mean data filtered on an individual retrieval-basis to remove known sources of remote-sensing bias. The results agree with the relationship derived from monthly mean data in McCoy et al. (2017a). Based on this

relationship, a first indirect radiative forcing of -0.97 Wm$^{-2}$ was derived. The forcing found in McCoy et al. (2017a) based on the stratocumulus regions and confirmed globally by this study is stronger than found in previous empirical remote-sensing studies (Bellouin et al., 2013;Quaas et al., 2008), but not out of line with climate model studies forced to be consistent with in-situ relationships between sulfate and CDNC (Storelvmo et al., 2009). Therefore remotely-sensed CCN proxies that are not speciated are not as skillful a predictor of true CCN variability as sulfate mass, and will underestimate the radiative

forcing due to aerosol-cloud interactions. This result also reinforces the validity of a strongly negative first indirect effect from other, idealized, estimates (Stevens, 2015).

In addition to showing the sensitivity of CDNC to SO$_4$, we have shown that submicron sea spray as predicted by MERRA2 does not strongly affect CDNC except at very low sea-spray mass. As noted above, the submicron sea-spray in MERRA2 is effectively dependent on wind speed, so the precise coefficient relating sea salt to CDNC should change

depending on the size distribution assumed by a different parameterization of sea spray emission, but should maintain the same qualitative dependence.





In summary, when remote-sensing retrieval biases are accounted for carefully, sulfate mass concentration near the surface covaries with observed CDNC in the same way in highly pristine and in polluted regions. Inter-annual variability in CDNC near passively degassing volcanoes agrees with both reanalysis $SO_4$ and observed $SO_2$. Further, the decadal trend in CDNC predicted by reanalysis aerosol in regions of maritime outflow near sources of intense anthropogenic pollution agrees

with observed trends in CDNC. This shows that the relation between CDNC and $SO_4$ has relevance to aerosol-cloud radiative forcing. To our knowledge this is the first study to use remote-sensing SO2 and CDNC to show that inter-annual variability in sulfur emissions alters CDNC. Based on this we suggest that the relation between sulfate mass and CDNC provides a constraint on aerosol-cloud interactions in GCMs.

**Author contributions**

DTM and FAMB planned the paper. DTM performed data analysis and calculations and wrote the text. DPG created the CDNC level 3 data set from level 2 MODIS data. JKM downloaded and organized the MERRA2 aerosol data. All coauthors edited the manuscript, discussed results, and supported analysis.

**Acknowledgments**

The MODIS data were obtained from NASA's Level 1 and Atmosphere Archive and Distribution System (LAADS),

https://ladsweb.modaps.eosdis.nasa.gov/. MERRA2 data was downloaded from the Giovanni data server. DTM and PRF acknowledge support from the PRIMAVERA project, funded by the European Union's Horizon 2020 programme, Grant Agreement no. 641727.

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





|  | SO4 | DU | BC | OC | SS | b | land-sea | Lon | Lat | r |
|---|---|---|---|---|---|---|---|---|---|---|
| **Peruvian** | 0.3 | 0.09 | -0.06 | 0.04 | -0.15 | 1.7 | Ocean | 115W-65W | 30S-0N | 0.64 |
| **Namibian** | 0.21 | 0.14 | 0.38 | -0.32 | -0.12 | 2.27 | Ocean | 20W-20E | 30S-0N | 0.61 |
| **Australian** | 0.44 | 0.09 | 0.01 | -0.03 | -0.12 | 1.94 | Ocean | 55E-120E | 35S-15S | 0.57 |
| **Californian** | 0.2 | 0 | -0.03 | 0.13 | -0.04 | 2.03 | Ocean | 150W-110W | 10N-40N | 0.43 |
| **Canarian** | 0.29 | 0.07 | -0.08 | 0.11 | -0.06 | 1.95 | Ocean | 40W-5W | 10N-40N | 0.53 |
| **China** | 0.27 | 0.01 | 0.01 | 0.05 | -0.02 | 2.11 | Ocean | 100E-160E | 10N-40N | 0.63 |
| **North Atlantic** | 0.24 | -0.03 | 0.15 | -0.03 | -0.07 | 2.03 | Ocean | 60W-0E | 40N-70N | 0.45 |
| **North East Pacific** | 0.08 | -0.04 | 0.07 | 0.03 | -0.02 | 1.96 | Ocean | 180W-120W | 40N-70N | 0.24 |
| **North West Pacific** | 0.11 | -0.05 | 0.18 | -0.04 | -0.01 | 2.17 | Ocean | 120E-180E | 40N-70N | 0.35 |
| **South East Pacific** | 0.29 | 0.09 | -0.14 | 0.02 | -0.1 | 1.64 | Ocean | 180W-70W | 70S-30S | 0.45 |
| **South Atlantic** | 0.29 | 0.03 | -0.09 | 0.04 | -0.11 | 1.73 | Ocean | 70W-60E | 70S-30S | 0.38 |
| **South Indian Ocean** | 0.3 | -0.02 | -0.04 | 0.01 | -0.07 | 1.81 | Ocean | 60E-180E | 70S-35S | 0.37 |
| **Galapagos** | 0.09 | 0.06 | 0.1 | -0.04 | 0 | 2.25 | Ocean | 120W-70W | 0N-10N | 0.37 |
| **Chinese Stratus** | 0.27 | 0.03 | -0.16 | 0.11 | -0.02 | 2.05 | Land | 100E-130E | 10N-40N | 0.42 |
| **Amazon** | 0.22 | 0 | 0.06 | -0.03 | 0.01 | 2.37 | Land | 80W-30W | 15S-10N | 0.46 |
| **Equatorial Africa** | 0.06 | -0.02 | 0.01 | 0.07 | 0.07 | 2.41 | Land | 20W-20E | 15S-15N | 0.37 |
| **North America** | 0.18 | 0.02 | 0.2 | -0.13 | 0.02 | 2.55 | Land | 100W-75W | 30N-45N | 0.33 |
| **India** | -0.02 | -0.01 | 0.39 | -0.24 | 0.05 | 2.71 | Land | 65E-90E | 10N-30N | 0.41 |
| **Europe** | 0.18 | 0.02 | 0.08 | 0 | 0.02 | 2.42 | Land | 0E-50E | 25N-45N | 0.37 |

**Table 1 Details of the regions considered in this study (see also Figure 1). For each region the coefficients relating CDNC to predictors from equation 1 are shown along with the correlation coefficient of the regression model in that region. The constant term in the regression is shown under b. The latitude-longitude bounding box of each region is shown and it is noted if data is restricted to being over land or oceans.**



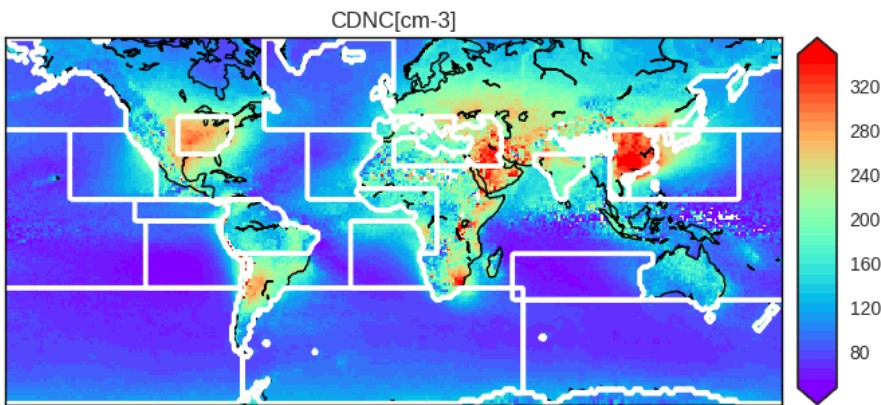

**Figure 1  The mean CDNC from 2003-2015 observed by MODIS. Boxes over land and ocean used to examine different regimes in Figure 3 are shown in white.**



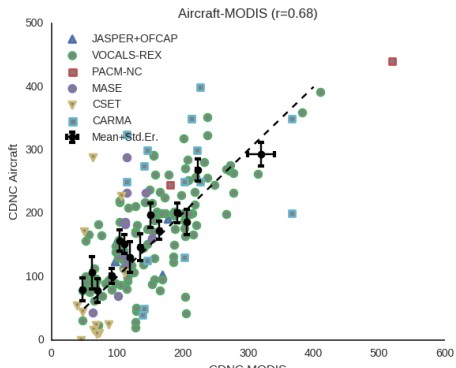

**Figure 2 Aircraft-measured CDNC versus MODIS CDNC where MODIS CDNC has been averaged within 1.5 degrees of the flight leg and 1.5 days. Data from JASPER and OFCAP near the Antarctic peninsula (Lachlan-Cope et al., 2016); VOCALS-REX off the coast of Peru (Allen et al., 2011); MASE, CSET, and CARMA off the California coast (Lu et al., 2007;Hegg et al., 2007); PACM-NC refers to data from data from Northern China near Beijing and Tianjin (Ma et al., 2010). The one to one relation is shown as a dashed line and the mean of the data is shown with black dots taken over equal quantiles of the data and the standard error in the mean is shown with error bars.**



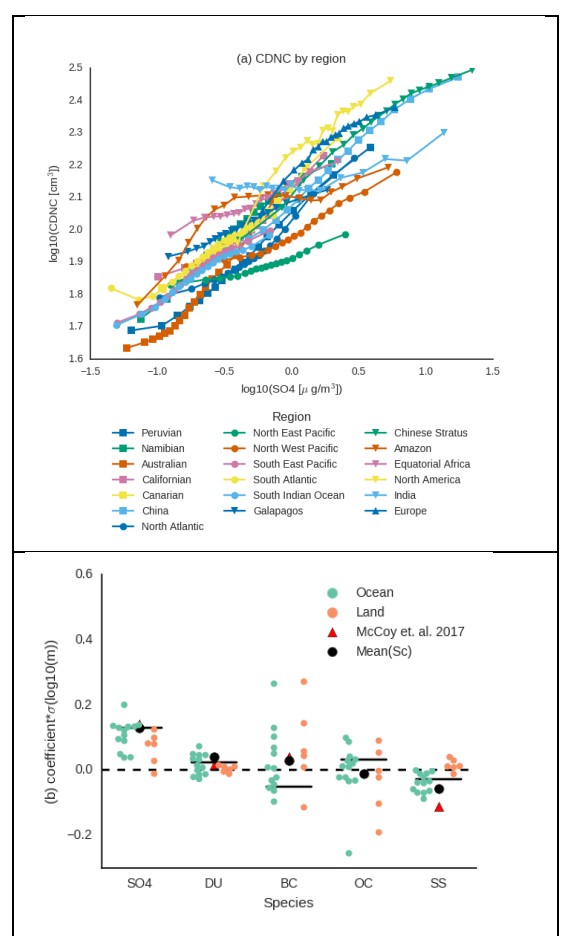

**Figure 3 (a) Daily CDNC from MODIS binned by boundary layer sulfate mass from MERRA2. Regions (Figure 1) are noted in the legend. (b) The multiple linear regression coefficients relating CDNC to boundary layer aerosol mass concentrations. Each coefficient is scaled by the standard deviation of the relevant predictor in the regression model. Black lines show the coefficient values if all available data is used to train the regression model, triangles show equivalent values from McCoy et al. (2017a) derived from measurements over stratocumulus decks, black circles show the mean coefficient values from the present study restricted to stratocumulus decks. Note that BC and OC are the mass that is predicted by MERRA2 to be hydrophilic and sea salt (SS) and dust (DU) are the mass that is predicted to be submicron. These distinctions have been made to try and look at the most CCN-relevant aerosol mass in these species. Coefficients for each region are shown in Table 1.**





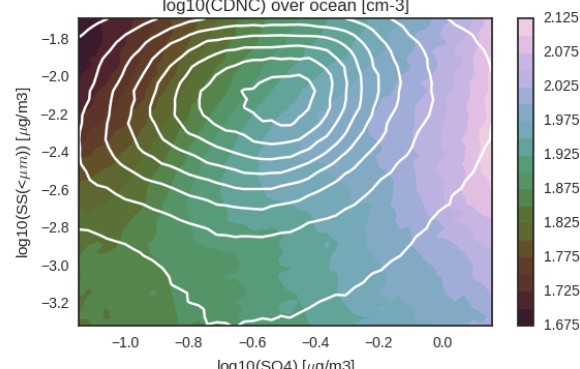

**Figure 4 The dependence of CDNC on submicron sea salt and sulfate mass predicted by MERRA2. All variables are shown in log$_{10}$-space. White lines show the probability distribution of sea salt and sulfate in the observations. Equivalent plots replacing sea salt with dust, black carbon, and organic carbon are shown in Figure S2.**



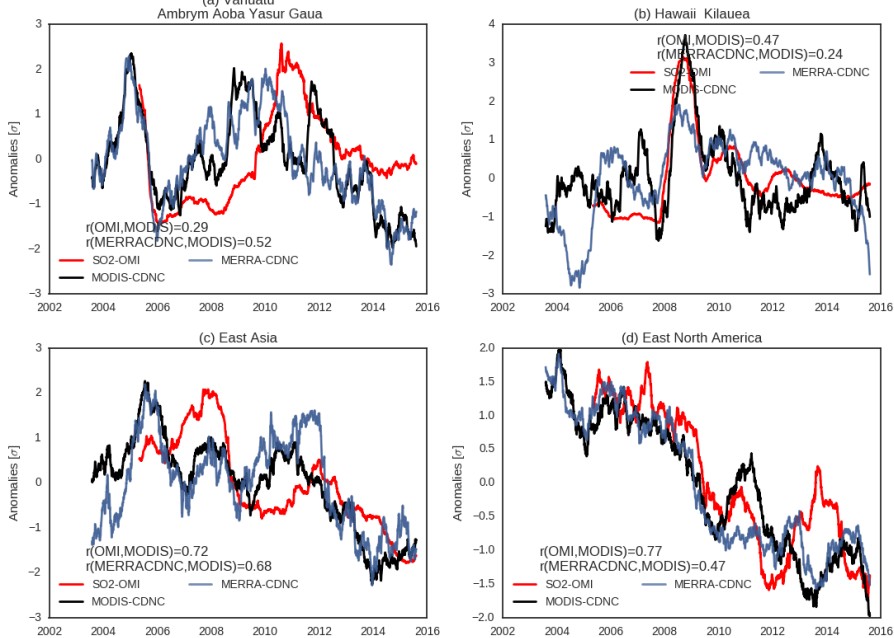

**Figure 5 Sulfur dioxide (SO2) from OMI, log$_{10}$ CDNC from MODIS, and log$_{10}$ CDNC predicted by MERRA2 SO4 over Vanuatu and Hawaii (a,b) and China and the east coast of the US (c,d). A 365-day running mean is used to smooth both time series. In (a) and (b) CDNC and SO2 are averaged within 5° of the volcano. In (c) and (d) SO2 is averaged over land and CDNC is averaged off the coast. All data are plotted in standard deviations relative to the mean of the time series. The correlation between time series of CDNC from MODIS and SO2 from OMI, and between predicted and observed CDNC are noted in the legend for each figure. Note that these correlations are taken before taking the 365-day running mean. The correlation between the time series of predicted and observed CDNC after taking the running mean is noted in Figure 6.**





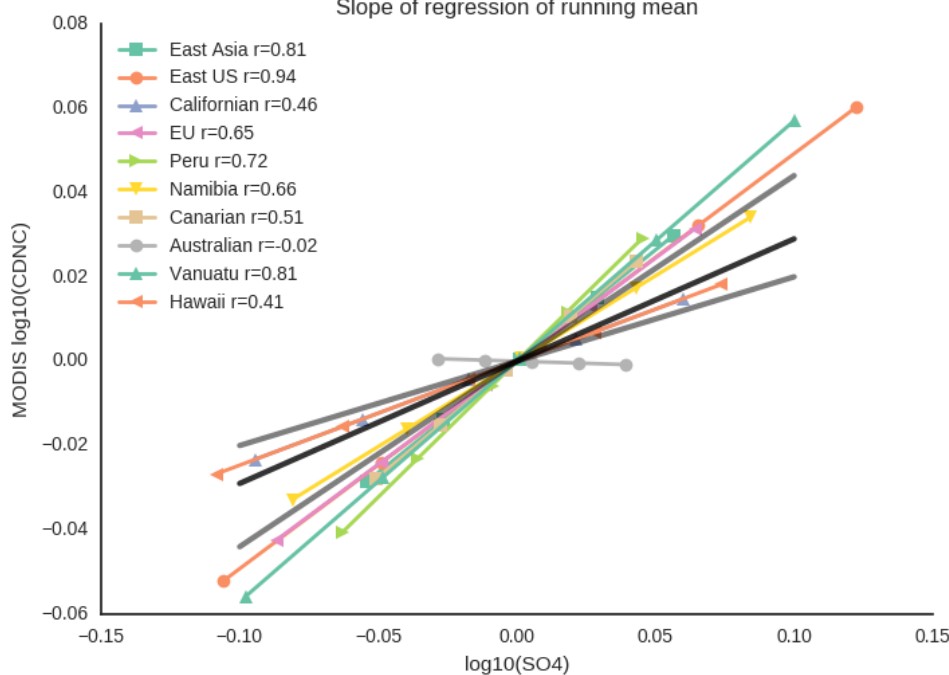

**Figure 6** An illustration of the slope of the linear regression of regionally and temporally averaged $\log_{10}SO4$ on $\log_{10}CDNC$. Both CDNC and SO4 are averaged in each region and smoothed with a 365-day moving average (see Figure 5). Each region is noted in the legend along with the correlation between $\log_{10}CDNC$ and $\log_{10}SO4$. The x-range of the lines corresponds to the range of regionally- and temporally-averaged $\log_{10}SO4$ in each region. Slopes derived from 1°x1° daily data in the stratocumulus regions are shown using black and grey solid lines. The mean coefficient from the stratocumulus regions (Table 1) is shown as a black line. The minimum and maximum coefficients from the stratocumulus regions (Californian and Australian, respectively) are shown as solid grey lines.