# Peer review of "Predicting decadal trends in cloud droplet number concentration using reanalysis and satellite data"

_Atmospheric Chemistry and Physics, 2017_

## Referee Comment (RC1) · Anonymous Referee #1 · 15 Oct 2017

This is a fascinating paper. It is remarkable that significant relationships between sulfur and droplet number are found for daily and interannual time scales. I have just a few minor comments.

Minor comments

Page 2, Line 22. Replace "effective" with "numerous". Larger particles are more effective in the sense that smaller supersaturations are needed to activate them.

Page 3, line 7. Are these in-cloud values, weighted by cloud fraction?

[Figure]

2017.

---

## Referee Comment (RC2) · Anonymous Referee #2 · 1 Nov 2017

General comments

In this paper, the authors used the filtered MODIS level 2 swath data which considered only low liquid clouds (cloud tops below 3.2 km) to calculate the daily-mean CDNC at 1°x1° resolution with the approach introduced in their previous work (McCoy et al., JGR, 2017). Then they validated their CDNC date set by comparing with aircraft measurements from the Antarctic Peninsula, Northern China, and the Peruvian and Californian stratocumulus decks. The comparison shown that the correlation between aircraft and satellite observations can be high up to 0.68. With multiple linear regression between MODIS CDNC and MERRA2 reanalysis masses of sulfate, black carbon,

organic carbon, sea salt, and dust at 910 hPa model level, the authors found CDNC across many different regimes can be reproduced by a simple power law fit to near-surface sulfate, with smaller contributions from other aerosols. Their investigation also indicated that decadal time scale reduction of CDNC over Asia and North America is agreed with the reduction of OMI observed SO2 over the same regions due to emission controls. This paper is well organized and presented. It is a good application and expansion of the work of McCoy et al., JGR, 2017.

Special comments

Page1Line1: The authors filtered MODIS L2 data set with low liquid clouds assumption. Therefore, 'low cloud droplet number concentration' is better than just say 'cloud droplet number concentration'.

Page2Line14-16: How do the authors average these daily time scale data set to the multi-year mean as shown in Figure 1?

Page2Line26-27: Could the authors provide brief descriptions on how the effective radius and optical depth values are retrieved by MODIS.

Page3Line5: Do the values of cloud fraction in MERRA2 also exceed 80% at the grids where MODIS cloud fractions exceed 80%?

Page3Line8-10: Is MODIS AOD used for nudging MERRA2 aerosol emission or mass loading?

Page4Line16-19: Validation of MODIS CDNC with aircraft measurement is important. Can the authors provide the data set of aircraft measurement employed in this study? What are the major differences of CDNC from McCoy et al., JGR, 2017 and CDNC from Bennartz and Rausch, ACP, 2017 comparing to aircraft measurement?

Page5Line26-27: As discussed by the authors previously, the coefficients in table 1 can be varied at different regions due to different atmospheric chemistry and physics processes. It is well known that atmospheric chemistry and physics processes also

be impacted by seasonal changes of emission and atmospheric environment. How do these factors impact the authors' conclusion?

Page7Line8: Please add space between 'OMI' and '('.

Page7Line16-17: There are numbers of significant disagreements of OMI SO2 peaks and MODIS CDNC peaks. More detailed discussions and explanations are requested here.

---

## Author Comment (AC1) · 23 Dec 2017

The comment was uploaded in the form of a supplement:
https://www.atmos-chem-phys-discuss.net/acp-2017-811/acp-2017-811-AC1-supplement.pdf

---

## Author Response (AR1)

**Response to reviewers: Predicting decadal trends in cloud droplet number concentration using reanalysis and satellite data**

Daniel T. McCoy[1*], Frida A.-M. Bender[2], Daniel P. Grosvenor[1], Johannes K. Mohrmann[3], Dennis L. Hartmann[3], Robert Wood[3], Paul R. Field[1,4]
[1]University of Leeds, Leeds, LS2 9JT, UK,
[2]Stockholm University, Stockholm, 114 18, Sweden
[3]University of Washington, Seattle, 98195, USA [4]MetOffice, Exeter, EX1 3PB, UK
*Correspondence to*: Daniel T. McCoy (d.t.mccoy@leeds.ac.uk)

We thank the reviewers for their consideration of our manuscript and for their helpful and supportive comments. Reviewer comments are in bold.

**REVIEWER 1**

**This is a fascinating paper. It is remarkable that significant relationships between sulfur and droplet number are found for daily and interannual time scales. I have just a few minor comments.**

Thank you- we are hopeful that this relationship will be helpful to people looking at aerosol-cloud interactions.

**Minor comments**

**Page 2, Line 22. Replace "effective" with "numerous". Larger particles are more effec- tive in the sense that smaller supersaturations are needed to activate them.**

Done. Thanks.

**Page 3, line 7. Are these in-cloud values, weighted by cloud fraction?**

They are in-cloud. Good point. We added a comment explaining this.

**REVIEWER 2**

**General comments**

**In this paper, the authors used the filtered MODIS level 2 swath data which considered only low liquid clouds (cloud tops below 3.2 km) to calculate the daily-mean CDNC at $1°x1°$ resolution with the approach introduced in their previous work (McCoy et al., JGR, 2017). Then they validated their CDNC date set by comparing with aircraft mea- surements from the Antarctic Peninsula, Northern China, and the Peruvian and Cal- ifornian stratocumulus decks. The comparison shown that the correlation between aircraft and satellite observations can be high up to 0.68. With multiple linear regres- sion between**

**MODIS CDNC and MERRA2 reanalysis masses of sulfate, black carbon, organic carbon, sea salt, and dust at 910 hPa model level, the authors found CDNC across many different regimes can be reproduced by a simple power law fit to near- surface sulfate, with smaller contributions from other aerosols. Their investigation also indicated that decadal time scale reduction of CDNC over Asia and North America is agreed with the reduction of OMI observed SO2 over the same regions due to emis- sion controls. This paper is well organized and presented. It is a good application and expansion of the work of McCoy et al., JGR, 2017.**

Thank you- we appreciate your help and your time looking over our paper.

**Special comments**

**Page1Line1: The authors filtered MODIS L2 data set with low liquid clouds assumption. Therefore, 'low cloud droplet number concentration' is better than just say 'cloud droplet number concentration'.**

This is a good point, but we feel that this is discussed in the text extensively and would prefer to not over-complicate the title. We have added a note to the abstract explaining that they are low altitude, liquid-topped clouds.

**Page2Line14-16: How do the authors average these daily time scale data set to the multi-year mean as shown in Figure 1?**

The average is not weighted so it does potentially alias in the seasonal cycle. For instance, CDNC retrievals are not possible in the midlatitudes during local winter so those parts of the map would be empty if we weighted it by the seasonal cycle. It is intended that the figure be illustrative of the existing data set, and is not intended as a climatology. We did not express this well. We have added verbiage to explain the intent of this figure. Thanks.

**Page2Line26-27: Could the authors provide brief descriptions on how the effective radius and optical depth values are retrieved by MODIS.**

We have now added text to explain how MODIS retrievals of effective radius and optical depths are performed. The paragraph also gives further details on how the data is filtered after the MODIS retrievals. It now reads :-

"In this paper and in McCoy et al. (2017a) CDNC is calculated from MODIS effective radius ($r_e$) and optical depth ($\tau$) retrievals using the adiabatic cloud assumption following Grosvenor and Wood (2014). MODIS simultaneously retrieves $\tau$ and $r_e$ via a bispectral algorithm that uses

reflectances from both a non-absorbing visible wavelength (0.65 μm over land and 0.86 μm over the ocean) and an absorbing shortwave infrared wavelength (either 1.6, 2.1 and 3.7 μm; Nakajima and King, 1990). In McCoy et al. (2017a) 1°x1° daily-mean MODIS ($r_e$) and ($\tau$) values were used to calculate CDNC, which were then averaged to monthly resolution. The use of this CDNC dataset may be problematic in some regions for a number of reasons (also see McCoy et al. (2017a)) : 1) it is subject to high solar zenith angle biases in the individual swaths, which were averaged together to create each daily data point; 2) biases may be present due to the use of area averaged $r_e$ and $\tau$ rather than using pixel level values for the CDNC calculation; 3) the dataset was not filtered to include low altitude clouds only, which may have led to a lack of connectivity between surface aerosol sources and cloud CDNC; 4) the CDNC was calculated using the 2.1μm MODIS channel $r_e$, which is likely to be affected more strongly by cloud heterogeneity related biases than the 3.7μm channel (Zhang et al., 2012).

In the present study, level-2 swath data (joint product) from MODIS collection 5.1 (King et al., 2003) is filtered to remove problematic retrievals at a pixel-level following Grosvenor and Wood (2014), including the removal of pixels with a solar zenith angle greater than 65$^o$. The daily-mean CDNC at 1°x1° resolution is calculated using filtered level 2 swath data and only low (cloud tops below 3.2 km), liquid clouds were used to calculate CDNC. Only 1°x1° regions where the cloud fraction exceeds 80% are considered valid (Bennartz et al., 2011) and the CDNC is calculated using the 3.7μm MODIS channel $r_e$. In the remainder of this paper MERRA2 data is only considered for days and 1°x1° regions when and where MODIS is able to perform a retrieval of CDNC within this set of criterion. Notably, in the comparison between predicted and observed CDNC in Section 3.2 and in the comparison of long-term trends in Section 3.3. The mean CDNC over the period 2003-2015 is shown in Figure 1. It should be noted Figure 1 is intended to illustrate the mean CDNC over the existing data set and CDNC values are not weighted to equally represent the seasonal cycle, for example in midlatitude winter retrievals of CDNC are not possible and these areas would be blank in a climatologically-weighted map. Values of CDNC are in-cloud."

**Page3Line5: Do the values of cloud fraction in MERRA2 also exceed 80% at the grids where MODIS cloud fractions exceed 80%?**

Good point- otherwise we might be aliasing in variability in where the retrieval is performed that is driven by clouds or the seasonal cycle. We sample MERRA2 and MODIS data in the same way when we examine the covariability in daily 1°x1° data and when we look at the regionally-averaged trends. We have added a note to this effect. Thank you.

**Page3Line8-10: Is MODIS AOD used for nudging MERRA2 aerosol emission or mass loading?**

For the mass loading- good catch (Randles et al., 2016;Buchard et al., 2015)- thanks. We have added more explanation.

**Page4Line16-19: Validation of MODIS CDNC with aircraft measurement is important. Can the authors provide the data set of aircraft measurement employed in this study? What are the major differences of CDNC from McCoy et al., JGR, 2017 and CDNC from Bennartz and Rausch, ACP, 2017 comparing to aircraft measurement?**

Our study differs in that we are investigating whether the population mean agrees between in-situ and remotely-sensed CDNC as opposed to the more intensive validation performed in Bennartz and Rausch. We discuss this in the methodology. It is not possible to directly compare the two studies given their much different methodologies. We also clearly state that the goal of this analysis is not to perform an exhaustive validation of the remotely-sensed CDNC with in-situ data, which is beyond the scope of the present study given that it would require the collection and compilation of an extensive database of in-situ observations. We just used published mean CDNC from flight legs provided in the text of the literature we cite. Our hope is that this analysis will provide support for the utility of creating a GASSP-like database of CDNC. We have added additional discussion of the differences between this analysis and previous papers and the source of the in-situ data. Thank you for this comment.

**Page5Line26-27: As discussed by the authors previously, the coefficients in table 1 can be varied at different regions due to different atmospheric chemistry and physics processes. It is well known that atmospheric chemistry and physics processes also be impacted by seasonal changes of emission and atmospheric environment. How do these factors impact the authors' conclusion?**

We find remarkable similarity across regions and regimes. This is regardless of whether the aerosol is strongly seasonal from biogenic sources or has relatively little seasonal variability because they are anthropogenically-controlled. We also span a wide variety of meteorological regimes. Overall this seems to point toward seasonal variability not playing a controlling role in determining the relationship between CDNC and aerosol mass concentrations. We have added discussion to this effect- good point, thanks.

**Page7Line8: Please add space between 'OMI' and '('.**

Thanks-done.

**Page7Line16-17: There are numbers of significant disagreements of OMI SO2 peaks and MODIS CDNC peaks. More detailed discussions and explanations are requested here.**

We only really expect the SO2 and CDNC to roughly agree and we discuss the limitations in remotely-sensed SO2. For example, eruptions that inject significant SO2 into the free troposphere have been manually removed here as best as we can (as discussed in the text) – further, in the examination of anthropogenic trends we average SO2 over a large continental region and the CDNC in the marine outflow so it seems likely that the vagaries of oxidative capacity and circulation will introduce slight disagreement in these quantities. We provide correlations between CDNC and SO2 without the removal of seasonal variability by use of the running mean. Overall the correlation is quite high in an objective sense, despite these departures. We have added discussion as to the expectation of dissimilarities at the end of section 3.3. We also provide the correlation coefficients between the time series and discuss limitations in the data sets. Thank you.

Buchard, V., da Silva, A. M., Colarco, P. R., Darmenov, A., Randles, C. A., Govindaraju, R., Torres, O., Campbell, J., and Spurr, R.: Using the OMI aerosol index and absorption aerosol optical depth to evaluate the NASA MERRA Aerosol Reanalysis, Atmos. Chem. Phys., 15, 5743-5760, 10.5194/acp-15-5743-2015, 2015.
Randles, C., AM, d. S., V, B., A, D., PR, C., V, A., H, B., EP, N., X, P., A, S., H, Y., and R, G.: The MERRA-2 Aerosol Assimilation, Technical Report Series on Global Modeling and Data Assimilation, 45, 2016.

[revised manuscript text omitted]

In this paper and in McCoy et al. (2017a) CDNC is calculated from MODIS effective radius ($r_e$) and optical depth ($\tau$) retrievals using the adiabatic cloud assumption following Grosvenor and Wood (2014). MODIS simultaneously retrieves

30    $\tau$ and $r_e$ via a bispectral algorithm that uses reflectances from both a non-absorbing visible wavelength (0.65 μm over land and 0.86 μm over the ocean) and an absorbing shortwave infrared wavelength (either 1.6, 2.1 and 3.7 μm; Nakajima and King (1990)). In McCoy et al. (2017a) 1°x1° daily-mean MODIS ($r_e$) and ($\tau$) values were used to calculate CDNC, which

were then averaged to monthly resolution. The use of this CDNC dataset may be problematic in some regions for a number of reasons (also see McCoy et al. (2017a)) : 1) it is subject to high solar zenith angle biases in the individual swaths, which were averaged together to create each daily data point; 2) biases may be present due to the use of area averaged $r_e$ and $\tau$ rather than using pixel level values for the CDNC calculation; 3) the dataset was not filtered to include low altitude clouds only, which may have led to a lack of connectivity between surface aerosol sources and cloud CDNC; 4) the CDNC was calculated using the 2.1μm MODIS channel $r_e$, which is likely to be affected more strongly by cloud heterogeneity related biases than the 3.7μm channel (Zhang et al., 2012).

In the present study, level-2 swath data (joint product) from MODIS collection 5.1 (King et al., 2003) is filtered to remove problematic retrievals at a pixel-level following Grosvenor and Wood (2014), including the removal of pixels with a solar zenith angle greater than 65°. The daily-mean CDNC at 1°x1° resolution is calculated using filtered level 2 swath data and only low (cloud tops below 3.2 km), liquid clouds were used to calculate CDNC. Only 1°x1° regions where the cloud fraction exceeds 80% are considered valid (Bennartz et al., 2011) and the CDNC is calculated using the 3.7μm MODIS channel $r_e$. In the remainder of this paper MERRA2 data is only considered for days and 1°x1° regions when and where MODIS is able to perform a retrieval of CDNC within this set of criteria, notably, in the comparison between predicted and observed CDNC in Section 3.2 and in the comparison of long-term trends in Section 3.3. The mean CDNC over the period 2003-2015 is shown in Figure 1. It should be noted that Figure 1 is intended to illustrate the mean CDNC over the existing data set and CDNC values are not weighted to equally represent the seasonal cycle, for example in midlatitude winter retrievals of CDNC are not possible and these areas would be blank in a climatologically-weighted map. Values of CDNC are retrieved only when a cloud is present and are thus in-cloud values and are not the average of cloud-free and cloudy regions.

[revised manuscript text omitted]

Mace, G. G., and Avey, S.: Seasonal Variability of Warm Boundary Layer Cloud and Precipitation Properties in the Southern Ocean as Diagnosed from A-Train Data, Journal of Geophysical Research: Atmospheres, n/a-n/a, 10.1002/2016JD025348, 2016.

Malavelle, F. F., Haywood, J. M., Jones, A., Gettelman, A., Clarisse, L., Bauduin, S., Allan, R. P., Karset, I. H. H., Kristjánsson, J. E., Oreopoulos, L., Cho, N., Lee, D., Bellouin, N., Boucher, O., Grosvenor, D. P., Carslaw, K. S., Dhomse, S., Mann, G. W., Schmidt, A., Coe, H., Hartley, M. E., Dalvi, M., Hill, A. A., Johnson, B. T., Johnson, C. E., Knight, J. R., O'Connor, F. M., Partridge, D. G., Stier, P., Myhre, G., Platnick, S., Stephens, G. L., Takahashi, H., and Thordarson, T.: Strong constraints on aerosol–cloud interactions from volcanic eruptions, Nature, 546, 485-491, 10.1038/nature22974

http://www.nature.com/nature/journal/v546/n7659/abs/nature22974.html - supplementary-information, 2017.

Matsui, T., Masunaga, H., Kreidenweis, S. M., Pielke, R. A., Tao, W. K., Chin, M., and Kaufman, Y. J.: Satellite-based assessment of marine low cloud variability associated with aerosol, atmospheric stability, and the diurnal cycle, Journal of Geophysical Research-Atmospheres, 111, D17204 10.1029/2005jd006097, 2006.

McCoy, D. T., Burrows, S. M., Wood, R., Grosvenor, D. P., Elliott, S. M., Ma, P.-L., Rasch, P. J., and Hartmann, D. L.: Natural aerosols explain seasonal and spatial patterns of Southern Ocean cloud albedo, Science Advances, 1, 2015.

McCoy, D. T., and Hartmann, D. L.: Observations of a substantial cloud-aerosol indirect effect during the 2014–2015 Bárðarbunga-Veiðivötn fissure eruption in Iceland, Geophys. Res. Lett., n/a-n/a, 10.1002/2015GL067070, 2015.

McCoy, D. T., Bender, F. A. M., Mohrmann, J. K., Hartmann, D. L., Wood, R., and Grosvenor, D. P.: The global aerosol-cloud first indirect effect estimated using MODIS, MERRA and AeroCom, Journal of Geophysical Research: Atmospheres, n/a-n/a, 10.1002/2016JD026141, 2017a.

McCoy, D. T., Field, P. R., Schmidt, A., Grosvenor, D. P., Bender, F. A. M., Shipway, B. J., Hill, A. A., and Wilkinson, J. M.: The aerosol-cyclone indirect effect in observations and high-resolution simulations, Atmospheric Chemistry and Physics, 2017b.

Meskhidze, N., and Nenes, A.: Phytoplankton and Cloudiness in the Southern Ocean, Science, 314, 1419-1423, 10.1126/science.1131779, 2006.

Mijling, B., van der A, R. J., Boersma, K. F., Van Roozendael, M., De Smedt, I., and Kelder, H. M.: Reductions of NO2 detected from space during the 2008 Beijing Olympic Games, Geophysical Research Letters, 36, n/a-n/a, 10.1029/2009GL038943, 2009.

Nakajima, T., and King, M. D.: Determination of the optical thickness and effective particle radius of clouds from reflected solar radiation measurements. Part I: Theory, Journal of the atmospheric sciences, 47, 1878-1893, 1990.

[revised manuscript text omitted]